# Dynamics and Concealment of Time-Delay Signature in Mutually Coupled Nano-Laser Chaotic Systems

Xueting Zhang [1], Gang Guo [2], Xintian Liu [2], Guosi Hu [1], Kun Wang [1] and Penghua Mu [1,*]

1   School of Physics and Electronic Information, Yantai University, Yantai 264005, China;
    zhangxueting0315@163.com (X.Z.); hugs@ytu.edu.cn (G.H.); wangkun2021@s.ytu.edu.cn (K.W.)
2   FISEC Infomation Technology Company Limited, Weihai 264200, China; gg87@fisherman-it.com (G.G.);
    liuxintian@fisherman-it.com (X.L.)
*   Correspondence: ph_mu@ytu.edu.cn

**Abstract:** It is well known that nano-lasers (NLs), as important optical components, have attracted widespread attention for their output characteristics. In this paper, the dynamic behavior and time-delay concealment properties of NLs mutually coupled in open-loop, semi-open-loop, and closed-loop structures have been numerically investigated. We employ bifurcation diagrams and 0–1 chaos tests in our simulations to quantitatively analyze the dynamic properties of the system and introduce the autocorrelation function to evaluate the ability of the system to conceal the time-delay signature (TDS). In the meantime, the effects of the NL parameters and the controllable variables of the system on the TDS are studied. The results indicate that, compared with an open-loop structure without feedback, the mutual coupling scheme with added feedback is beneficial for the system to output high-quality chaotic signals. Furthermore, selecting a moderate Purcell factor $F$ and a smaller spontaneous emission coupling factor $\beta$ can achieve TDS concealment over a wider parameter range of injection intensity and frequency detuning.

**Keywords:** nano-lasers; chaos; time-delay signatures





## 1. Introduction

The dynamic behavior of semiconductor lasers (SLs) has received much attention in the past few decades. When subjected to external perturbations, a variety of complex nonlinear properties are exhibited by SLs [1–3]. Among these, the generated chaotic light, characterized by noise-like properties, unpredictability, and a wide bandwidth, has found extensive applications in fields such as chaotic secure communications [4–8], high-speed random number generation [9,10], neural computing [11,12], reservoir computing [13], compressive sensing [14], and laser radar [15,16]. It is widely accepted that external perturbations include optical feedback [17], optical injection [18], optoelectronic feedback [19], frequency-selective feedback [20], and current modulation [21], etc. However, the external cavity structure inevitably introduces periodicity in SLs systems, which is referred to as a time-delay signature (TDS) [22]. This periodicity in the signal can degrade the performance of chaos-based applications. For example, the presence of time-delay information poses a risk of information leakage, where, by extracting the time-delay information, eavesdroppers can obtain the key parameters of the communication system and reconstruct the chaotic secure system, thereby threatening the security of chaotic secure communication systems [23]. At the same time, the introduction of time-delay features also reduces the random performance of random number generators [9]. Consequently, concealing the TDS within the system is a focal point of interest for researchers.

Currently, there are several major time series analysis techniques used to extract the time delay, including the autocorrelation function (ACF) [22], permutation entropy (PE) [24], delayed mutual information (DMI) [25], and others. In recent years, various approaches to reduce or even eliminate TDSs [26–35] have been proposed by a number

of studies. For example, introducing multiple feedback loops into the system (which involves considering complex feedback schemes), adding another laser, or incorporating multiple injection paths. Semiconductor nano-lasers (NLs) have attracted considerable interest due to their compact size, low power consumption, and promising potential in photonic integrated circuits (PICs) [36,37]. Hence, in recent years, the application of this type of laser in chaotic systems has been extensively investigated by a multitude of researchers. They have proposed several theoretical models of innovative semiconductor NLs to explore nonlinear dynamics [38–52] and TDS concealment [53–55]. In these systems, the dynamic properties of NLs are significantly influenced by two crucial parameters: the Purcell factor $F$ and the spontaneous emission coupling factor $\beta$ [53]. For instance, Ding et al. investigated the impact of $F$ and $\beta$ on the performance of electrically pumped NLs [41]. In addition, Sattar et al. conducted a comprehensive investigation of the dynamic behavior of NLs in different schemes, including conventional optical feedback [42], optical injection [43], and phase-conjugate feedback [45]. These studies detail the pathways through which chaos generates and the effects of internal and external parameters on the output of the system. Interestingly, it was observed that when the values of $F$ and $\beta$ are relatively high, NLs require a higher feedback rate to generate chaos. This is because larger values of $F$ and $\beta$ enhance the damping of the relaxation oscillations of NLs. Han et al. further explored the dynamic characteristics of mutually coupled semiconductor NLs and observed rich dynamic output. There, attention was given to the role played by $F$ and $\beta$ with different distances, D, between the NLs and for a range of NL bias currents [46–49]. Elsonbaty et al. studied the TDS concealment of semiconductor NLs using a hybrid all-optical and electro-optical feedback scheme. At the same time, they harnessed the generated chaotic light source for image encryption [50]. Xiang et al. found that the output from an NL, subjected to dual chaotic injection from two main NLs, exhibits a low TDS over a wide parameter range [53]. In previous work, we investigated TDS concealment in an unidirectional injection system and subsequently achieved secure communication based upon it [54]. Moreover, we extended our exploration by delving into the nonlinear dynamics of NLs under the influence of distributed feedback from fiber Bragg gratings (FBG). This investigation involved the modification of rate equations, incorporating variables such as $F$ and $\beta$. Our findings revealed that employing FBG feedback surpasses mirror feedback in its ability to conceal TDSs and expand effective bandwidth, especially when selecting intermediate feedback strength and frequency detuning, as described in reference [55]. Therefore, these studies have provided evidence that NLs, similar to traditional SLs, exhibit an extensive array of dynamic characteristics. However, it is worth noting that only a few studies have explored mutually coupled NLs and the mechanisms of TDS concealment in NLs have not been fully investigated. In particular, the influence of $F$ and $\beta$ on TDS concealment remains to be discussed. Thus, in this situation, it is appropriate to study the TDS of mutual coupling NLs systems to determine whether such systems can offer new functionalities in the context of PICs.

In this paper, we investigate the dynamic behavior and TDS concealment of mutual coupling NLs systems and discuss the impact of parameters on TDS concealment. The organization of this paper is as follows: In Section 2, the open-loop model of the mutually coupled NLs (without feedback) as well as the semi-open-loop model (with one feedback) and the closed-loop model (with two feedbacks) formed by the addition of the feedbacks are presented in detail. We also present the rate equations and parameter definitions corresponding to these three models. In Section 3, firstly we introduce the dynamic characteristics of NLs in the open-loop system and their TDS concealment, and then we study the TDSs in the semi-open-loop and closed-loop systems, focusing on the effects of $F$ and $\beta$ on TDS concealment. Finally, we draw fundamental conclusions based on our findings in Section 4.

## 2. Nano-Laser Dynamics

The structures of the mutual coupled NL systems that were studied are shown in Figure 1, which consists of three components. Figure 1a shows an open-loop system consisting of two NLs, namely the NL1 (the first NL) and the NL2 (the second NL), which are mutually injected without any external feedback. Figure 1b illustrates a semi-open-loop structure, which is an extension of the open-loop system with an additional feedback mechanism. In this configuration, NL1 receives optical feedback and then injects NL2. Figure 1c shows a closed-loop structure where both NL1 and NL2 receive optical feedback and mutually inject.

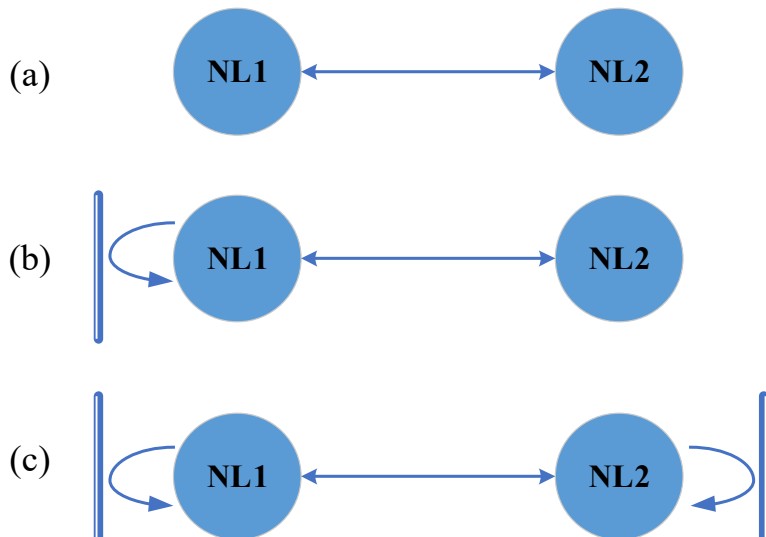

**Figure 1.** Schematic diagram of mutually coupled NLs. (**a**) An open-loop structure; (**b**) a semi-open-loop structure; and (**c**) a closed-loop structure. NL1: the first nano-laser, NL2: the second nano-laser.

The control parameters in this system include the feedback parameters, the injection parameters, the bias current *I*, *F*, and *β*. Our primary focus will be to investigate the impact of injection parameters (injection strength and frequency detuning) as well as the parameters *F* and *β* on the TDS. The rate equations for the three proposed systems are as follows [49,50]:

$$
\frac{dS_{1,2}(t)}{dt} = \Gamma\left[\frac{F\beta N_{1,2}(t)}{\tau_n} + \frac{g_n(N_{1,2}(t)-N_0)}{1+\varepsilon S_{1,2}(t)}\right] - \frac{1}{\tau_p}S_{1,2}(t)
$$
$$
+2kr_{1,2}\sqrt{S_{1,2}(t)S_{2,1}(t-tr)}\cos(\theta_{1,2}(t)) + 2kd_{1,2}\sqrt{S_{1,2}(t)S_{1,2}(t-td)}\cos(\theta_{3,4}(t))
\tag{1}
$$

$$
\frac{dN_{1,2}(t)}{dt} = \frac{I_{1,2}}{eV_b} - \frac{N_{1,2}(t)}{\tau_n}(F\beta + (1-\beta)) - \frac{g_n(N_{1,2}(t)-N_0)}{1+\varepsilon S_{1,2}(t)}S_{1,2}(t)
\tag{2}
$$

$$
\frac{d\phi_{1,2}(t)}{dt} = \frac{\alpha}{2}\Gamma g_n(N_{1,2}(t)-Nth) \pm \Delta\omega - kr_{1,2}\sqrt{\frac{S_{2,1}(t-tr)}{S_{1,2}(t)}}\sin(\theta_{1,2}(t))
$$
$$
-kd_{1,2}\sqrt{\frac{S_{1,2}(t-td)}{S_{1,2}(t)}}\sin(\theta_{3,4}(t))
\tag{3}
$$

$$
\theta_{1,2}(t) = \pm\Delta\omega t + 2\pi f_{2,1}tr + \phi_{1,2}(t) - \phi_{2,1}(t-tr)
\tag{4}
$$

$$
\theta_{3,4}(t) = 2\pi f_{1,2}td + \phi_{1,2}(t) - \phi_{1,2}(t-td)
\tag{5}
$$

where the subscripts "1" and "2" denote NL1 and NL2, respectively. $S(t)$ is the photon density, $N(t)$ is the carrier density, and $\phi(t)$ is the phase. $F$ is the Purcell factor, $\beta$ is the

spontaneous emission coupling factor, $\Gamma$ is the confinement factor, $g_n$ is the differential gain, $\varepsilon$ is the gain saturation factor, and $\alpha$ is the linewidth enhancement factor. $\tau_n$ and $\tau_p$ are the carrier lifetime and photon lifetime, respectively. $N_0$ is the transparent carrier density, $Nth(Nth = N_0 + 1/\Gamma\tau_p g_n)$ is the threshold carrier density. $e$ is the electron charge, $V_b$ is the volume of the active region. $I(I = 2Ith)$ is the bias current, where $Ith$ is the threshold current. $\Delta\omega(\Delta\omega = 2\pi\Delta f, \Delta f = f_1 - f_2)$ represents the angular frequency detuning between NL1 and NL2, where $\Delta f$ is the frequency detuning.

Note that the last term in Equations (1) and (3) stands for the feedback term, which contains the feedback delay $td$ and the feedback strength $kd$, and $kd$ can be expressed as [53]:

$$kd = f(1 - R)\sqrt{\frac{R_{ext}}{R}}\frac{c}{2nL} \tag{6}$$

where $f$ denotes the feedback coefficient, $R$ is the cavity surface reflectivity of NLs, $R_{ext}$ is the mirror reflectivity, $c$ is the speed of light, $n$ is the refractive index, and $L$ is the length of the feedback cavity. Similarly, in Equations (1) and (3), the penultimate term represents the injection term, where $kr$ and $tr$ are the injection strength and injection delay of the injection path, respectively. Here, $R_{inj}$ is the injection ratio and its calculation formula is defined as [54]:

$$kr = (1 - R)\sqrt{\frac{R_{inj}}{R}}\frac{c}{2nL} \tag{7}$$

We numerically solve Equations (1)–(5) using the fourth-order Runge–Kutta method in this study. Then, the noise term is ignored in the simulation. Table 1 lists the values of some important parameters used.

**Table 1.** Parameters used in numerical simulations [54].

| Parameter | Description | Value |
|-----------|-------------|-------|
| $\Gamma$ | Confinement factor | 0.645 |
| $\tau_n$ | Carrier lifetime | 1 ns |
| $\tau_p$ | Photon lifetime | 0.36 ps |
| $tr$ | Injection delay | 0.05 ns |
| $td$ | Feedback delay | 0.2 ns |
| $g_n$ | Differential gain | $1.64 \times 10^{-6}$ cm$^3$/s |
| $N_0$ | Transparency carrier density | $1.1 \times 10^{-18}$ cm$^{-3}$ |
| $\varepsilon$ | Gain saturation factor | $2.3 \times 10^{-17}$ cm$^3$ |
| $\alpha$ | Linewidth enhancement factor | 5 |
| $V_b$ | Volume of active region | $3.96 \times 10^{-13}$ cm$^3$ |
| $\lambda_0$ | Wavelength of NL | 1591 nm |
| $R$ | Laser facet reflectivity | 0.85 |
| $R_{ext}$ | External facet power reflectivity | 0.95 |
| $R_{inj}$ | Injection ratio | 0–0.1 |
| $n$ | Refractive index | 3.4 |
| $L$ | Cavity length | 1.39 μm |
| $Q$ | Quality factor | 428 |
| $f$ | Feedback coupling fraction | 0–0.9 |

To quantify the corresponding TDS, we employed the autocorrelation function (ACF), and in the following study, the TDS is considered to be concealed when the peak of the ACF is less than 0.2. The ACF is defined as [54]:

$$C(\Delta t) = \frac{\langle[I(t+\Delta t) - \langle I(t+\Delta t)\rangle][I(t) - \langle I(t)\rangle]\rangle}{\sqrt{\langle[I(t+\Delta t) - \langle I(t+\Delta t)\rangle]^2\rangle\langle[I(t) - \langle I(t)\rangle]^2\rangle}} \tag{8}$$

where $\langle\rangle$ represents the average value of the time series and $\Delta t$ represents the lag time.

## 3. Results and Discussion

In this section, we present the simulation results of the TDSs generated by the open-loop, semi-open-loop and closed-loop structures of the three systems. We focus on studying the effects of injection strength, frequency detuning, and the *F* and $\beta$ parameters on TDS concealment.

### 3.1. The Open-Loop Structure (without Feedback)

Firstly, we consider the nonlinear dynamics and TDS characteristics of an open-loop system, which is an NL system without optical feedback. Figure 2 shows the photon density time series plots of NL1 and NL2 at different injection strengths, where *F* = 14, $\beta$ = 0.05, *I* = 2*Ith*, and *Ith* = 1.1 mA. As can be seen from Figure 2, when the injection strengths are 13, 16, 22, and 31 ns$^{-1}$, respectively, NL1 and NL2 sequentially exhibit steady-state, periodic, period-doubling, and chaotic dynamic characteristics. At the same time, we can also observe that the outputs of NL1 and NL2 are similar.

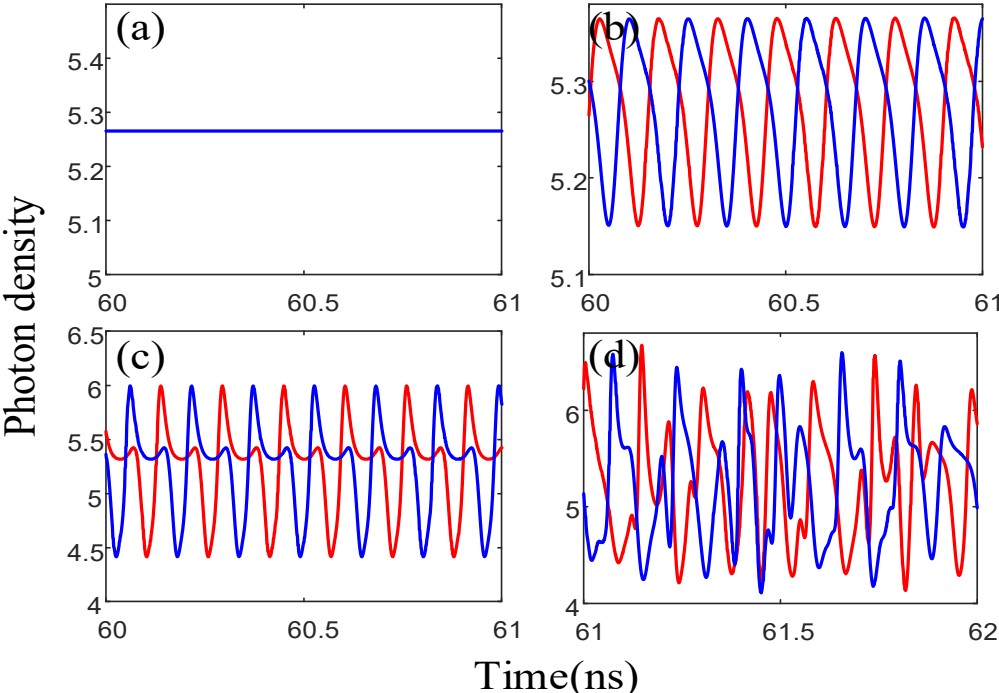

**Figure 2.** Photon density time series of NL1 and NL2. The injection strength *kr* values are (**a**) 13 ns$^{-1}$, (**b**) 16 ns$^{-1}$, (**c**) 22 ns$^{-1}$, and (**d**) 31 ns$^{-1}$, respectively. *F* = 14, $\beta$ = 0.05, *I* = 2*Ith*, *Ith* = 1.1 mA. The blue color represents NL1, and the red color represents NL2.

To further illustrate the evolutionary path of the dynamics, we use bifurcation diagrams of the photon density *S(t)* as a function of the injection intensity *kr*, as shown in Figure 3. Therefore, from Figure 3a, it can be seen that at an injection strength of *kr* < 15.9 ns$^{-1}$, the output is stable. When 15.9 < *kr* < 24.4 ns$^{-1}$ is reached, we can observe periodic and subharmonic phenomena in the plot, and chaos starts to occur around *kr* > 26.3 ns$^{-1}$. At this point, the frequency detuning between NL1 and NL2 is $\Delta f = 0$. Again, NL2 has similar bifurcation properties, as in Figure 3b.

The 0–1 test of chaos holds significant importance in chaos research as it enables the distinction between chaotic systems and those that are not. The dynamic output of the NLs can be easily controlled by varying the injection parameters and the frequency detuning. Figure 4 shows the chaos 0–1 test plot with frequency detuning on the horizontal axis and injection intensity on the vertical axis. The left plot corresponds to the 0–1 test plot for NL1, whereas the right plot corresponds to the 0–1 test plot for NL2. If the 0–1 test value is 1, this indicates that NLs produce chaotic output. Conversely, if the test value is any other value, it means that the NLs are in a non-chaotic output state. From Figure 4, it is clear that

NL1 and NL2 are more prone to generate chaos in the range of $-10\,\text{GHz} < \Delta f < 10\,\text{GHz}$ and $kr > 26\,\text{ns}^{-1}$. This finding is similar to those observed in Figures 2 and 3, indicating that the 0–1 chaos test plot matches well with the expected outcomes of the system.

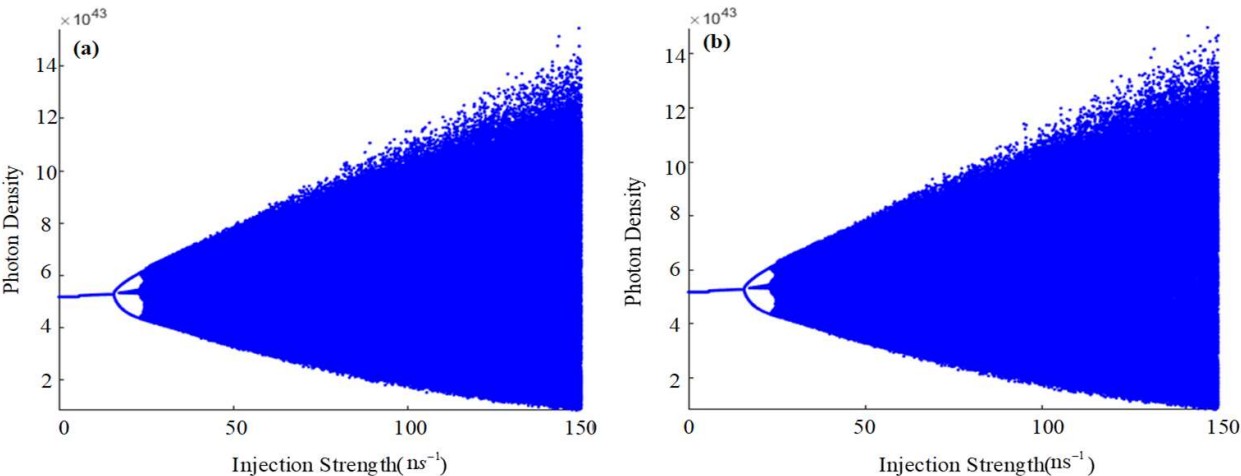

**Figure 3.** The bifurcation diagram of the photon density and injection intensity for (**a**) NL1 and (**b**) NL2 at $F = 14$, $\beta = 0.05$, $I = 2Ith$, and $Ith = 1.1$ mA.

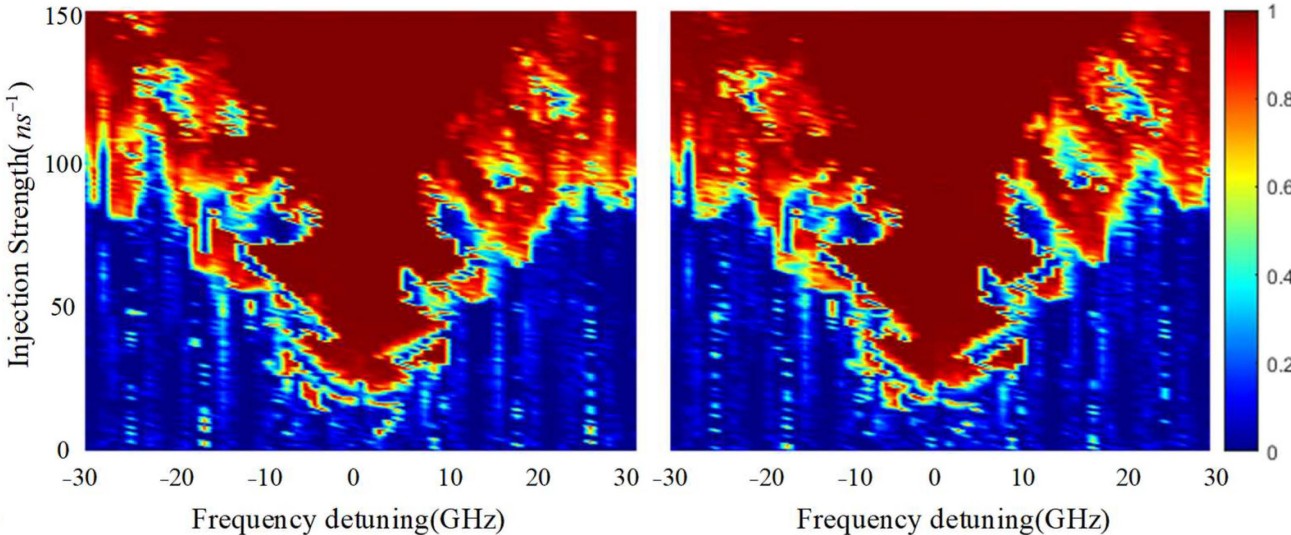

**Figure 4.** The 0–1 test for chaos of NL1 and NL2 in the parameter plane $(\Delta f, kr)$. $F = 14$, $\beta = 0.05$.

Next, we investigate the TDSs of the two NL outputs, with the parameters set as above. Figure 5 depicts the TDS and ACF plots for NL1 and NL2. It can be seen that both NL1 and NL2 are operating in a chaotic state, but they have distinct peaks, as shown in Figure 5. Meanwhile, Figure 6 displays the two-dimensional color plots of the TDSs for NL1 and NL2. The horizontal axis represents frequency detuning, whereas the vertical axis represents injection strength. In the plot, the regions with ACF values less than 0.2 are shown in deep blue, indicating that the TDSs in these regions are well concealed. For clarity, we use an ACF peak equal to 0.2 as a dividing line. It is worth noting that the lower a value, the higher the complexity of chaotic output and the better the time delay can be suppressed. Consequently, we designate the parameter region below 0.2 as $L$. From Figure 6, we can see that NL1 and NL2 have comparable TDS concealment capabilities and the $L$ region appears to have a "V" shape. More specifically, over the whole range of injection strength, the region where TDS can be concealed is relatively small. Therefore, to enhance the system's ability to conceal time delay, we propose incorporating feedback.

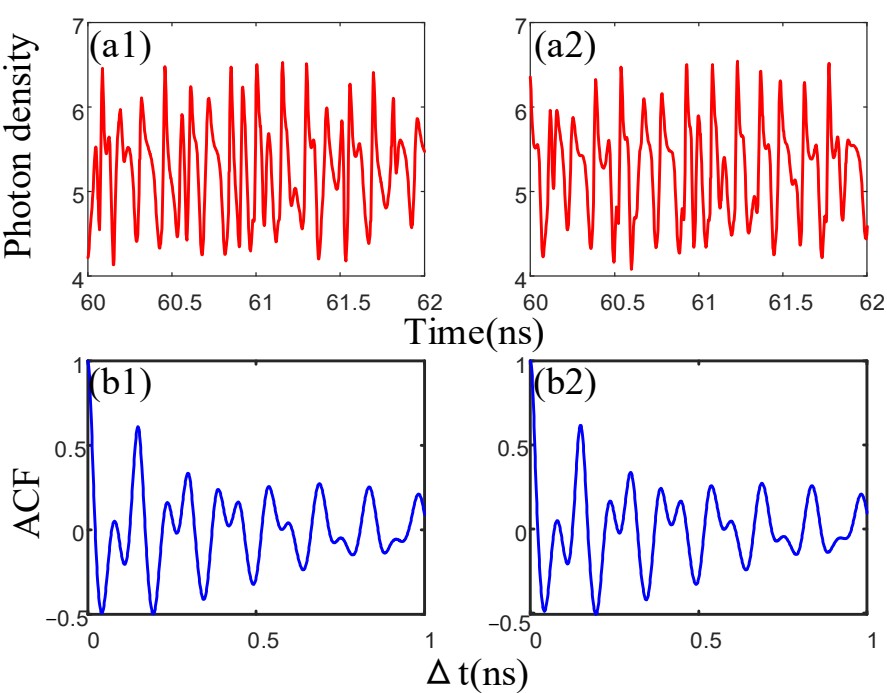

**Figure 5.** (**a1**,**a2**) Intensity time series for NL1 and NL2; (**b1**,**b2**) ACF. $F$ = 14, $\beta$ = 0.05, $I$ = $2Ith$, $Ith$ = 1.1 mA; injection strength $kr$ = 31 ns$^{-1}$.

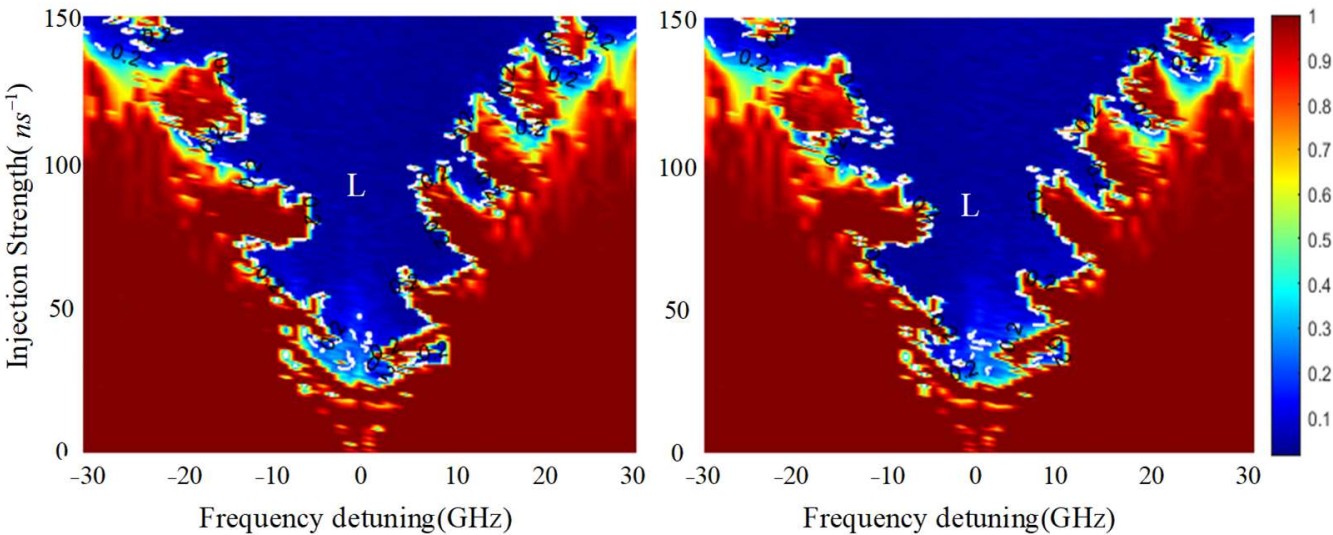

**Figure 6.** Two-D map of chaos of NL1 and NL2 in the parameter plane $(\Delta f, kr)$. $F$ = 14, $\beta$ = 0.05.

### 3.2. The Semi-Open-Loop Structure (with One Feedback)

In this section, we consider the addition of a feedback cavity to NL1 for investigation. The time series results and the corresponding ACF are shown in Figure 7. It can be distinctly observed from Figure 7(a1,a2) that both NL1 and NL2 operate in chaotic states and it is not immediately evident whether they exhibit significant TDSs. However, the relevant time-delay information extracted from the ACF diagram is visible, and from Figure 7(b1,b2), we can see that NL1 has a TDS greater than 0.2, whereas NL2 has a TDS less than 0.2. In this case, NL2 achieves TDS concealment, which is attributed to the chaotic output of NL1 subjected to optical feedback being injected into NL2, enabling NL2 to better conceal the TDS. Simultaneously, Figure 7(a1,a2) displays that the photon-density time series of NL1 has a slightly higher intensity than NL2 in the semi-open-loop structure. This is due to NL1 being subjected to external optical feedback.

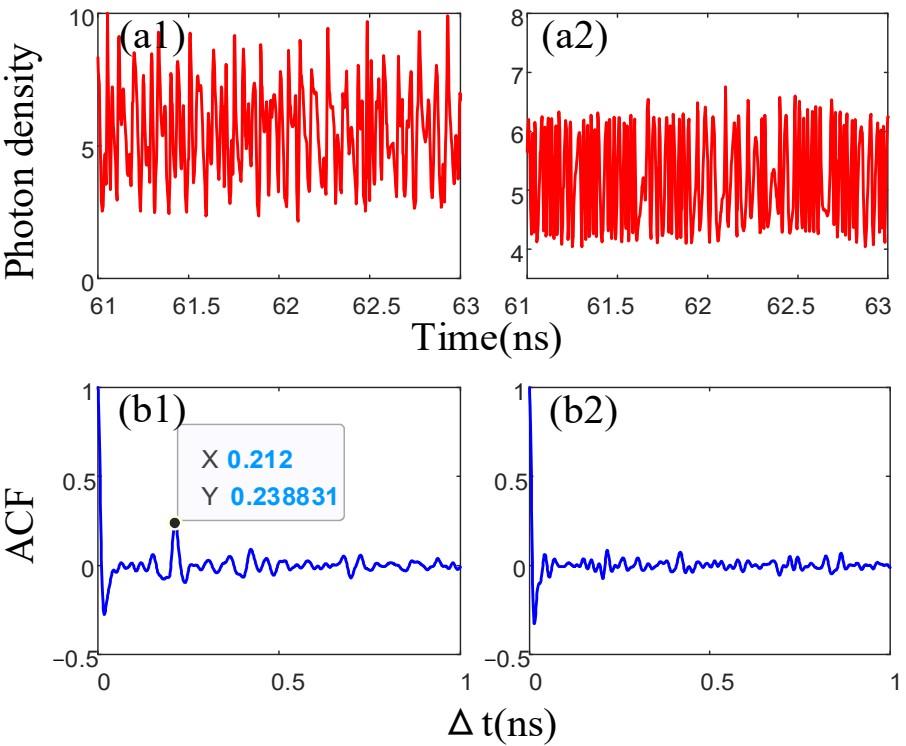

**Figure 7.** (**a1**,**a2**) Time series for NL1 and NL2; (**b1**,**b2**) ACF curve. $F = 14$, $\beta = 0.05$, $I = 2Ith$, $Ith = 1.1$ mA; feedback coefficient $f = 0.014$; injection strength $kr = 31$ ns$^{-1}$.

Previous studies have demonstrated that $F$, $\beta$, and bias current $I$ have a significant impact on the dynamics of NLs. It should be noted that in this study, we assume that the TDS is successfully suppressed when the ACF peak is less than 0.2. Firstly, the variation in TDS values for NLs with different $F$, $\beta$ and $I$ values was investigated; the results are shown in Figures 8 and 9. As a whole, the two curves (NL1 and NL2) exhibit the same tendency, i.e., the size of the ACF peak decreases and tends to stabilize as the injection strength $kr$ increases. Figure 8c,d show a relatively poor ability to conceal time delay, especially in (d), where $F = 30$ and $\beta = 0.1$. This is because the larger values of $F$ and $\beta$ enhance the damping of the relaxation oscillations of the NLs. However, as shown in Figure 8a,b, when $F = 10$ and $\beta = 0.05$, as well as when $F = 14$ and $\beta = 0.05$, the ACF peak values are consistently below 0.2 over the entire range of injection strengths, which achieves better time-delay concealment. Nevertheless, within the range of injection strength $kr < 40$ ns$^{-1}$, the value of ACF for NL2 is even lower, indicating that NL2 achieves more effective time-delay concealment, consistent with the results in Figure 7. Then, Figure 9 illustrates the results for $q = 2, 3, 4$, and 5 ($I = qIth$), which show both NL1 and NL2 can achieve TDS values below 0.2 in a larger range of injection strength, and the simulation result plots are extremely similar. This suggests that the effect of $I$ on the peak value of the ACF in the system is small. Therefore, $I = 2Ith$ is still selected for further investigation in subsequent experiments.

Subsequently, the effects of $F$ and $\beta$ on TDS concealment were further quantitatively evaluated. The two-dimensional mapping of the ACF peak sizes in the parameter space of frequency detuning and injection strength are shown in Figure 10, from which several features can be observed; as the values of $F$ and $\beta$ increase, the low peak region (dark blue region) shrinks. These results may be attributed to the fact that the increase in $F$ and $\beta$ results in stronger damping of the relaxation oscillations and the stronger damping of the relaxation oscillations leads to the better stability, which in turn decreases the hidden region of the TDS. In addition, for the sake of clarity, the region where the peak equals 0.2 is marked with contour lines. By comparing Figure 10(a1) with (b1), (a2) with (b2), and (a3) with (b3), it can be observed that the dark blue region of NL2 is larger than that of NL1 and that NL2 is larger in the range of $kr < 40$ ns$^{-1}$, which suggests that the TDS

concealment effect of NL2 is better, and the conclusion is consistent with that of Figures 7–9 in the experimental results. Therefore, in order to better conceal the TDS of NLs in a wider range of parameters, it is recommended to use NLs with moderate *F* and small *β* values.

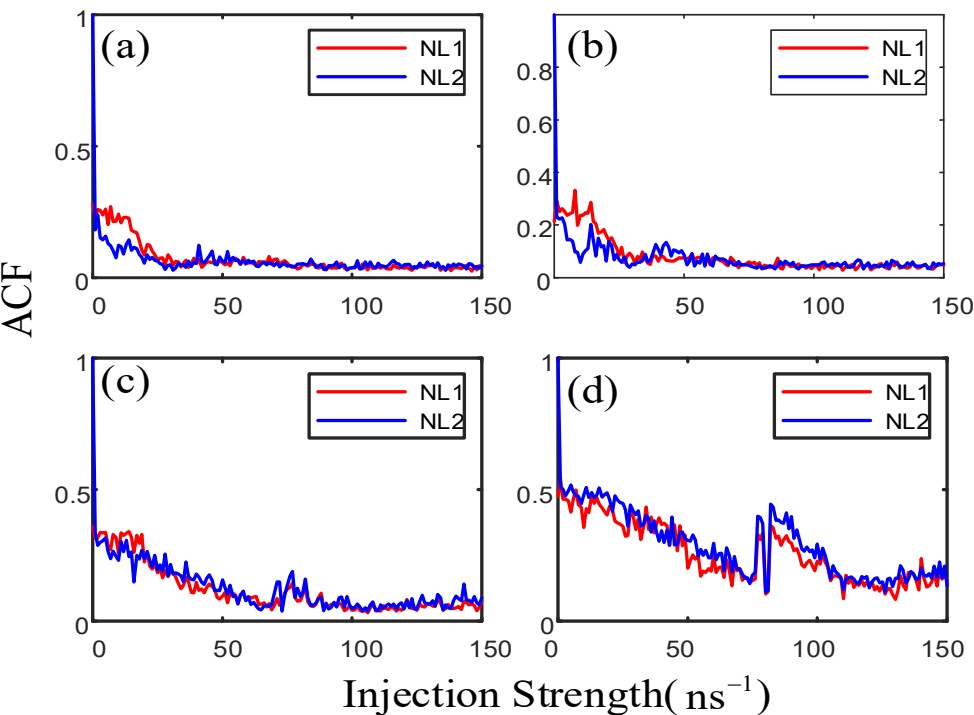

**Figure 8.** Relationship between ACF peak size and injection intensity in NL1 and NL2 at different *F* and *β* values. (**a**) *F* = 10, *β* = 0.05; (**b**) *F* = 14, *β* = 0.05; (**c**) *F* = 14, *β* = 0.1; (**d**) *F* = 30, *β* = 0.1; *I* = 2*Ith*, *Ith* = 1.1 mA.

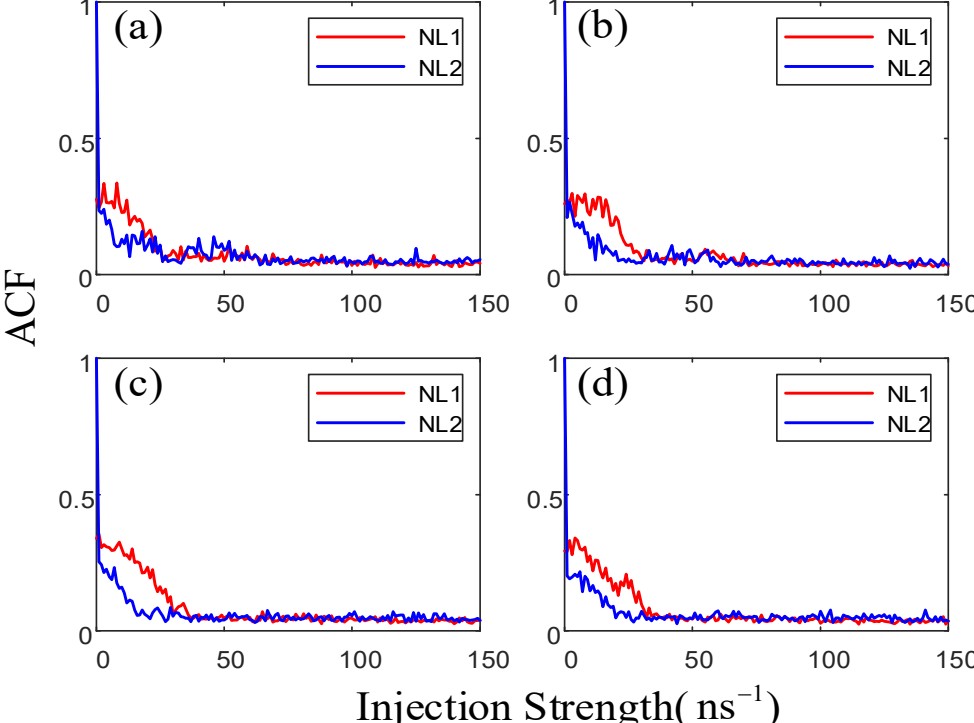

**Figure 9.** Relationship between ACF peak size and injection intensity in NL1 and NL2 at different bias currents *I* (*I* = *qIth*). (**a**) *q* = 2; (**b**) *q* = 3; (**c**) *q* = 4; (**d**) *q* = 5; *F* = 14, *β* = 0.05.

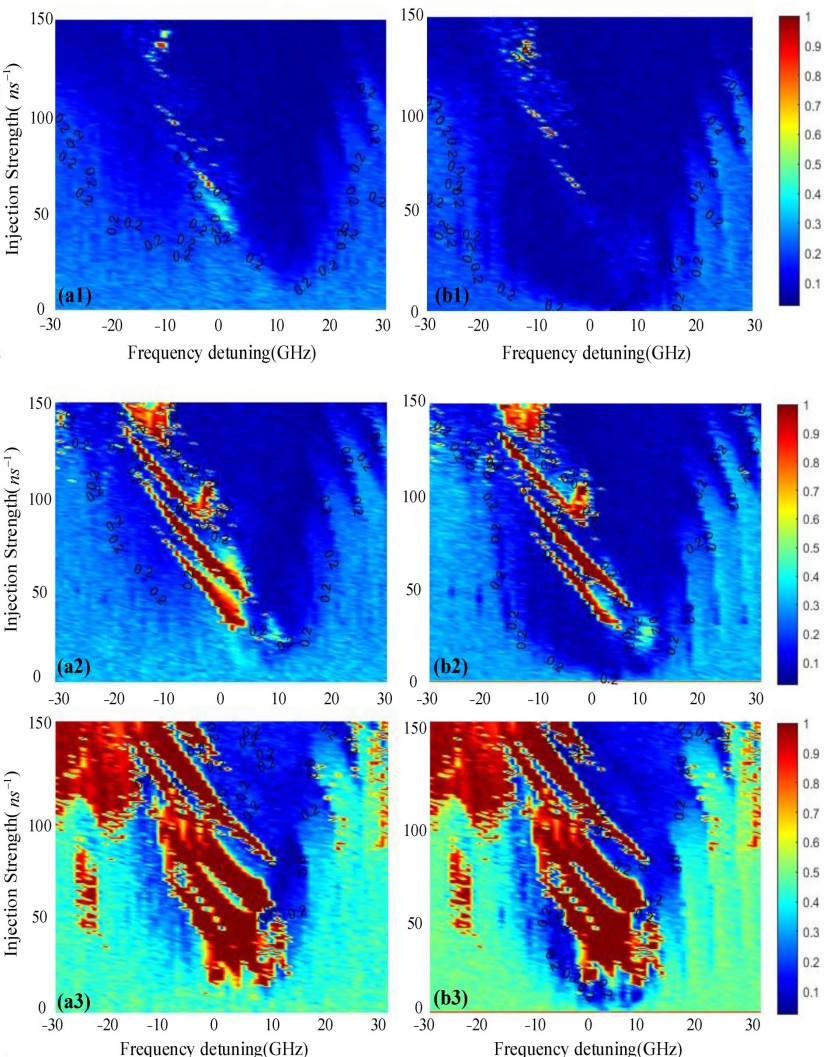

**Figure 10.** Two-dimensional color maps of the ACF of chaotic outputs from NL1 and NL2 as a function of frequency detuning and injection intensity variations under different *F* and *β*. *I = 2Ith*; (**a1–a3**) NL1; (**b1–b3**) NL2; (**a1,b1**) *F* = 14, *β* = 0.05; (**a2,b2**) *F* = 14, *β* = 0.1; (**a3,b3**) *F* = 30, *β* = 0.1.

### 3.3. The Closed-Loop Structure (with Two Feedbacks)

From the above research, it can be concluded that the semi-open-loop structure can effectively achieve TDS concealment compared with the open-loop structure without feedback. To further investigate the variation in the TDS in the mutually coupled system, we next study the closed-loop structure where both NL1 and NL2 are subjected to external optical feedback. In a similar way, the effects of *F* and *β* on TDS performance are also considered in this section. Figure 11 shows the intensity time series and ACF of NL1 and NL2 in the closed-loop structure system. It can be observed that both NL1 and NL2 operate in a chaotic state without TDSs. Figure 12 is a two-dimensional color map of the ACFs of NL1 and NL2 in the parameter space of the frequency detuning and injection strength. When *F* = 14 and *β* = 0.05, as shown in Figure 12(a1,b1), the maximum peak value in the entire region is 0.18, indicating that the NLs in the closed-loop structure achieve TDS concealment over the whole parameter plane. When *F* = 14 and *β* = 0.1, with *F* fixed and *β* slightly increased, the concealed region in Figure 12(a2,b2) is close to that in Figure 12(a1,b1). However, when *F* = 30 and *β* = 0.1, it can be observed from the bottom of Figure 12(a3,b3) that in a smaller range of injection strength, the two NLs cannot achieve TDS concealment. In this case, the results suggest this data has the worst performance in achieving TDS concealment among these three data, which is consistent with the results of the previous experiment. In fact, this is expected because the TDS in the semi-open loop structure

has already been well suppressed. Then, adding feedback under the same parameters introduces additional optical nonlinearity, which further increases the complexity of the output chaos. As a result, a greater range for time-delay concealment can be achieved.

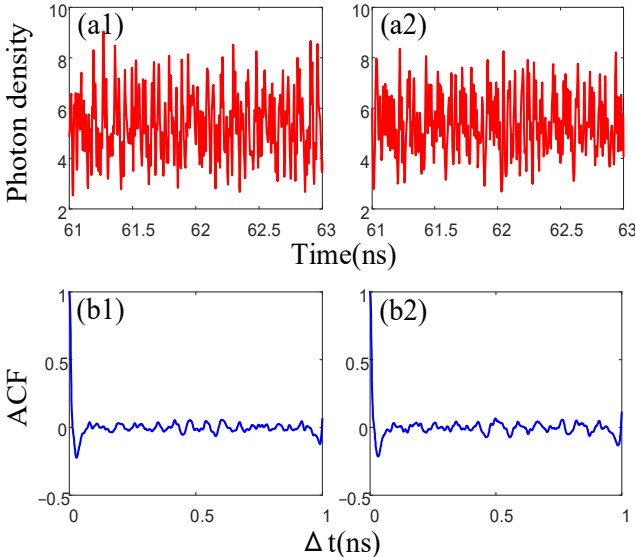

**Figure 11.** (**a1**,**a2**) The intensity time series of NL1 and NL2; (**b1**,**b2**) ACF. $F = 14$, $\beta = 0.05$, $I = 2Ith$, $Ith = 1.1$ mA; feedback coefficient $f = 0.01$; injection strength $kr = 31$ ns$^{-1}$.

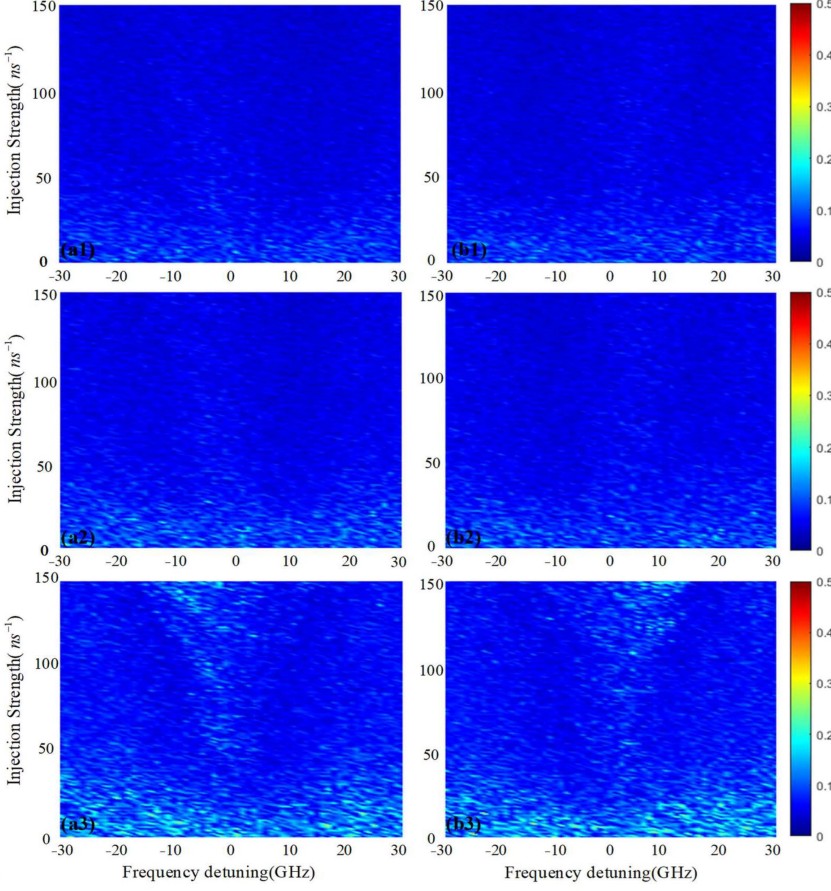

**Figure 12.** Two-dimensional color maps of the ACF of chaotic outputs from NL1 and NL2 as a function of frequency detuning and injection intensity variations under different $F$ and $\beta$. $I = 2Ith$; (**a1**–**a3**) NL1; (**b1**–**b3**) NL2; (**a1**,**b1**) $F = 14$, $\beta = 0.05$; (**a2**,**b2**) $F = 14$, $\beta = 0.1$; (**a3**,**b3**) $F = 30$, $\beta = 0.1$.

## 4. Conclusions

In this paper, based on the rate equations of NLs, we have numerically investigated the dynamic properties and TDS concealment of the open-loop, semi-open-loop, and closed-loop structures of mutually coupled NLs systems. In addition, the effects of the system parameters, the Purcell factor *F*, the spontaneous emission coupling factor *β*, and the bias current *I* on the TDS concealment of the chaotic outputs of the NLs are explored in detail. The numerical simulation results show that for the semi-open-loop structure, the effect of *I* on TDS is relatively small, whereas moderate *F* and smaller *β* values can achieve TDS concealment over a wider range of the injection strength and frequency detuning. Furthermore, it can be seen that the closed-loop structure, due to the addition of yet another feedback, introduces additional optical nonlinearity, which allows NLs to achieve TDS concealment in almost the entire parameter region. Therefore, our research confirms that adopting the coupling scheme with feedback can greatly enhance the dynamics of NLs, expand the region of TDS concealment, and provide a theoretical basis for chaos-based applications.

**Author Contributions:** Methodology, X.Z. and P.M.; validation, G.G. and P.M.; investigation, X.L., G.H. and K.W.; writing—original draft preparation, X.Z., G.G. and K.W.; writing—review and editing, X.Z., X.L., P.M. and K.W. All authors have read and agreed to the published version of the manuscript.

**Funding:** This work was supported by the Project: Natural Science Foundation of Shandong Provincial (ZR2020QF090), The Key Lab of Modern Optical Technologies of Education Ministry of China, Soochow University (KJS2066), and The Key Lab of Advanced Optical Manufacturing Technologies of Jiangsu Province, Soochow University (KJS2045).

**Institutional Review Board Statement:** Not applicable.

**Informed Consent Statement:** Not applicable.

**Data Availability Statement:** Not applicable.

**Acknowledgments:** The authors would like to thank all reviewers for their helpful comments and suggestions on this paper.

**Conflicts of Interest:** The authors declare no conflict of interest.

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
