# Peer review of "Dynamics and Concealment of Time-Delay Signature in Mutually Coupled Nano-Laser Chaotic Systems"

_photonics, doi:10.3390/photonics10111196_

Round 1

Reviewer 1 Report

In this manuscript, the authors numerically investigated the dynamic behavior and time-delay concealment of NLs mutually coupled system with open-loop, semi-open-loop, and closed-loop structures. By choosing the proper system parameters (injection strength and frequency detuning) and laser parameters (F and β) , the high quality chaos could be reached in the mutually coupled system with semi-open-loop and closed-loop structures. To my opinion, the presented theoretical work seems, overall, timely, of good quality and relevant in view of the current state of the art in the field. However, there are also several issues see my comments below - that needs to be addressed before publication of the article. I would therefore recommend publication after revision.

1)        The operating current is a very important parameter to study the output properties of NLs, and there are corresponding studies in many previous literatures, i.g., "Dynamics and Stability of Mutually Coupled Nano-Lasers,IEEE Journal of Quantum Electronics2016.11.205211):2000306 ". Therefore, the authors should take into account the impact of pumping current on the TDS concealment of this system.

2)        Figure 5 shows that NL1 and NL2 reach chaos synchronization, however, under the same injection strength in Figure 2(d), NL1 and NL2 do not achieve chaotic synchronization.

3)        There are some grammatical errors in the text. For example, P3 should be "The control parameters in this system include the feedback parameters, the injection parameters, the bias current I, F, and β."

4)        In Figures 8 and 9, there is a typographical error in the units of the horizontal axis.

5)        In Figure 8, the threshold current Ith=1.1mA when both F and β choose different values, to be my knowledge, the threshold current is dependent on the values of  F and β, the authors should check carefully the parameters in the simulation.

There are some grammatical errors in the text. 

Author Response

Reply: Thanks for supporting our work. Our reply to your questions and suggestions is detailed as follows.

1) The operating current is a very important parameter to study the output properties of NLs, and there are corresponding studies in many previous literatures, i.g., "Dynamics and Stability of Mutually Coupled Nano-Lasers, IEEE Journal of Quantum Electronics2016.11.205211):2000306 ". Therefore, the authors should take into account the impact of pumping current on the TDS concealment of this system.

Reply: Yes, we agree with the reviewer's perspective, and we also know that the operating current is an important parameter for the output of NLs. Therefore, we conducted a corresponding study on the semi-open-loop structure of mutually coupled NLs in this manuscript, as shown in Figure 9. The results show that the effect of the operating current I on the TDS is very small. Consequently, in the subsequent study, we will continue to employ the same operating current for our research.

2) Figure 5 shows that NL1 and NL2 reach chaos synchronization, however, under the same injection strength in Figure 2(d), NL1 and NL2 do not achieve chaotic synchronization.

Reply: Thanks. This problem is interesting. This is because in the procedure, we define the initial values of the photon density S(t), the carrier density N(t) and the phase Q(t) as random. Consequently, each output result will vary slightly, leading to the synchronization of NL1 and NL2 in Figure 5 at the same injection intensity, while Figure 2(d) did not synchronize.

3) There are some grammatical errors in the text. For example, P3 should be "The control parameters in this system include the feedback parameters, the injection parameters, the bias current I, F, and β."

Reply: We thank the reviewer for pointing out this. According to your suggestions, we have made some changes as follows:

 “The control parameters in this system include the feedback parameter, the injection parameter, the bias current I, F and β.” was modified to “The control parameters in this system include the feedback parameters, the injection parameters, the bias current I, F and β.”

4) In Figures 8 and 9, there is a typographical error in the units of the horizontal axis.

Reply: Thanks. We modified this in our revised manuscript.

5) In Figure 8, the threshold current Ith=1.1mA when both F and β choose different values, to be my knowledge, the threshold current is dependent on the values of F and β, the authors should check carefully the parameters in the simulation.

Reply: We greatly appreciate the reviewer for raising this point. However, existing studies have indicated that minimal fluctuations in the threshold current do not affect the general trends discovered during the computational process. Hence, we have used the threshold current Ith=1.1mA throughout the manuscript. This is substantiated in the following reference: "Enhancement of Nanolaser Time Delays Concealment and Unpredictability under External Field Manipulation, Acta Phys. Sin. Vol. 70, No. 11 (2021) 114201."

Reviewer 2 Report

The authors have made a useful study of the dynamics of nano-lasers. The specific topic is one which has been explored widely using other kinds of semiconductor lasers and it is for that reason that the originality, significance  and interest has been graded as average. For those with a focus on nano-lasers the contents of the paper would be of greater significance. The MS is very well prepared and the authors explain their work well.

The section of the paper dealing with the simulation results is termed ' Experimental results' which this referee considers inappropriate - most readers would expect such a section to contain results of device measurements not simulations. Amending that title would make the paper acceptable for publication.

Author Response

1The section of the paper dealing with the simulation results is termed ' Experimental results' which this referee considers inappropriate - most readers would expect such a section to contain results of device measurements not simulations. Amending that title would make the paper acceptable for publication.

Reply: Dear reviewer, thank you very much. According to your suggestions, this change have been made to the manuscript. Heading “Experimental results and Discussion” was amended to “Results and Discussion

Reviewer 3 Report

In the present article, the authors describe numerically and graphically the dynamic behavior and time-delay concealment properties of NLs mutually coupled in open-loop, semi-open-loop, and closed-loop structures.

The authors employ bifurcation diagrams and 0-1 chaos tests in their simulations to quantitatively analyze the dynamic properties of the system,

The authors introduce the autocorrelation function to evaluate the ability of the system to conceal the time delay signature (TDS).

The authors also discuss the effects of the NLs parameters and the controllable variables of the system on TDS.

Moreover, they mention that selecting a moderate Purcell factor F and a smaller spontaneous emission coupling factor β can achieve TDS concealment over a wider parameter range of injection intensity and frequency detuning.

This paper is written very well. Methodology adapted in this work is standard. The author describes rigorously numerical and graphical simulation.

The following are minor comments and suggestions:

w  Introduction: Enhance introduction by adding some recent references.

w  Page 2, Heading Section 2: “Theoretical Model” is not proper heading. Modelling has not been done in this section.

w  Figures: Increase the little sizes of Figures 2, 5, 7, 8, 9 and 11.

w  References: Mostly references are older than 5 years. Better to add at least 5 recent references from last 5 years and delete some of them.

The following are typos errors:

w  In Abstract, line 13: “we” should be “We”.  

w  Page 2: “section *” should be “Section *”

w  Page 4, line 122: “We” should be “we”.  

w  Page 4, line 128: “Where” should be “where”.  

w  Page 5, line 139: “the Figure 2” should be “Figure 2”.  

w  Page 5, line 140: “strength” should be “strengths”.  

w  Page 6, line 165: “Figure 2, 3” should be “Figures 2, 3”.

* Page 7, line 208: “Figure 8, 9” should be “Figures 8, 9”.

Little spelling errors

Author Response

Reply: Dear reviewer, thank you very much. According to your suggestions, several changes have been made to the manuscript.

The following are minor comments and suggestions:

Introduction: Enhance introduction by adding some recent references.

Reply: We thank the reviewer for pointing out this. According to your suggestions, we enhanced introduction by adding some references in the revised manuscript. There have few new references are added, so the reference mark is changed in the revised manuscript.

They proposed several theoretical models of innovative semiconductor NLs to explore nonlinear dynamics [38–52] and TDS concealment [53–55].

Han et al. further explored the dynamic characteristics of mutually coupled semiconductor NLs and observed rich dynamic output. There attention was given to the role played by F and β with different distances, D, between the NLs and for a range of NL bias currents [46–49]. Elsonbaty et al. studied the TDS concealment of semiconductor NLs using a hybrid all-optical and electro-optical feedback scheme. At the same time, they harnessed the generated chaotic light source for image encryption [50]. Xiang et al. found that the output from a NL, subjected to dual chaotic injection from two main NLs, exhibits low TDS over a wide parameter range [53]. In previous work, we investigated TDS concealment in an unidirectional injection system and subsequently achieved secure communication based on it [54]. Moreover, we extended our exploration by delving into the nonlinear dynamics of NLs under the influence of distributed feedback from fiber Bragg gratings (FBG). This investigation involved the modification of rate equations, incorporating variables such as F and β. Our findings revealed that employing FBG feedback surpasses mirror feedback in its ability to conceal TDS and expand effective bandwidth, especially when selecting intermediate feedback strength and frequency detuning, as described in reference [55].

Page 2, Heading Section 2: “Theoretical Model” is not proper heading. Modelling has not been done in this section.

Reply: We thank the reviewer for pointing out this. According to your suggestions, this change have been made to the manuscript. Heading “Theoretical Model” was amended to “Nano-laser Dynamics”.

Figures: Increase the little sizes of Figures 2, 5, 7, 8, 9 and 11.

Reply: Thanks. We modified this in our revised manuscript.

References: Mostly references are older than 5 years. Better to add at least 5 recent references from last 5 years and delete some of them.

Reply: Thanks. According to your suggestions, we added recent references from last 5 years and delete some of them in the revised manuscript.

4. “Argyris, A.; Syvridis, D.; Larger, L.; Annovazzi-Lodi, V.; Colet, P.; Fischer, I.; García-Ojalvo, J.; Mirasso, C.R.; Pesquera, L.; Shore, K.A. Chaos-Based Communications at High Bit Rates Using Commercial Fibre-Optic Links. Nature 2005, 438, 343–346, doi:10.1038/nature04275” was replaced by “Yan, S.-L. Chaotic laser parallel series synchronization and its repeater applications in secure communication. Acta Phys. Sin. 2019, 68, 170502, doi:10.7498/aps.68.20190212.”7. “Terry, J.R.; VanWiggeren, G.D. Chaotic Communication Using Generalized Synchronization. Chaos, Solitons & Fractals 2001, 12, 145–152, doi:10.1016/S0960-0779(00)00038-2.” was replaced by “Jiang, N.; Zhao, A.; Xue, C.; Tang, J.; Qiu, K. Physical Secure Optical Communication Based on Private Chaotic Spectral Phase Encryption/Decryption. Opt. Lett., OL 2019, 44, 1536–1539, doi:10.1364/OL.44.001536.

9. “Uchida, A.; Amano, K.; Inoue, M.; Hirano, K.; Naito, S.; Someya, H.; Oowada, I.; Kurashige, T.; Shiki, M.; Yoshimori, S.; et al. Fast Physical Random Bit Generation with Chaotic Semiconductor Lasers. Nature Photon 2008, 2, 728–732, doi:10.1038/nphoton.2008.227.” was replaced by “Li, P.; Sun, Y.; Liu, X.; Yi, X.; Zhang, J.; Guo, X.; Guo, Y.; Wang, Y. Fully Photonics-Based Physical Random Bit Generator. Opt. Lett., OL 2016, 41, 3347–3350, doi:10.1364/OL.41.003347.

15. “Lin, F.-Y.; Liu, J.-M. Chaotic Lidar. IEEE Journal of Selected Topics in Quantum Electronics 2004, 10, 991–997, doi:10.1109/JSTQE.2004.835296.” was replaced by Cheng, C.-H.; Chen, C.-Y.; Chen, J.-D.; Pan, D.-K.; Ting, K.-T.; Lin, F.-Y. 3D Pulsed Chaos Lidar System. Opt. Express, OE 2018, 26, 12230–12241, doi:10.1364/OE.26.012230.

16. “Lin, F.-Y.; Liu, J.-M. Chaotic Radar Using Nonlinear Laser Dynamics. IEEE Journal of Quantum Electronics 2004, 40, 815–820, doi:10.1109/JQE.2004.828237.” was replaced by “Chen, C.-Y.; Cheng, C.-H.; Pan, D.-K.; Lin, F.-Y. Experimental Generation and Analysis of Chaos-Modulated Pulses for Pulsed Chaos Lidar Applications Based on Gain-Switched Semiconductor Lasers Subject to Optical Feedback. Opt. Express, OE 2018, 26, 20851–20860, doi:10.1364/OE.26.020851.

17, “Lenstra, D.; Verbeek, B.; Den Boef, A. Coherence Collapse in Single-Mode Semiconductor Lasers Due to Optical Feedback. IEEE Journal of Quantum Electronics 1985, 21, 674–679, doi:10.1109/JQE.1985.1072725.” was replaced by “Liu, B.; Jiang, Y.; Ji, H. Sensing by Dynamics of Lasers with External Optical Feedback: A Review. Photonics 2022, 9, 450, doi:10.3390/photonics9070450.

18. “Kovanis, V.; Gavrielides, A.; Simpson, T.B.; Liu, J.M. Instabilities and Chaos in Optically Injected Semiconductor Lasers. Applied Physics Letters 1995, 67, 2780–2782, doi:10.1063/1.114591.” was replaced by “Yarunova, E.A.; Krents, A.A.; Molevich, N.E.; Anchikov, D.A. Suppression of Spatiotemporal Instabilities in Broad-Area Lasers with Pump Modulation by External Optical Injection. Bull. Lebedev Phys. Inst. 2021, 48, 55–58, doi:10.3103/S1068335621020081.

19. “Xia, G.-Q.; Chan, S.-C.; Liu, J.-M. Multistability in a Semiconductor Laser with Optoelectronic Feedback. Opt. Express, OE 2007, 15, 572–576, doi:10.1364/OE.15.000572.” was replaced by “Dmitriev, P.S.; Kovalev, A.V.; Locquet, A.; Citrin, D.S.; Viktorov, E.A.; Rontani, D. Predicting Chaotic Time Series Using Optoelectronic Feedback Laser. In Proceedings of the Semiconductor Lasers and Laser Dynamics IX; SPIE, April 1 2020; Vol. 11356, pp. 100–105.

20. “Wyatt, R. Spectral Linewidth of External Cavity Semiconductor Lasers with Strong, Frequency-Selective Feedback. Electronics Letters 1985, 15, 658–659, doi:10.1049/el:19850467.” was replaced by “Kaur, B.; Jana, S. Generation and Dynamics of One- and Two-Dimensional Cavity Solitons in a Vertical-Cavity Surface-Emitting Laser with a Saturable Absorber and Frequency-Selective Feedback. J. Opt. Soc. Am. B-Opt. Phys. 2017, 34, 1374–1385, doi:10.1364/JOSAB.34.001374.

This is an added reference:55.  Jiang, P.; Zhou, P.; Li, N.; Mu, P.; Li, X. Characterizing the Chaotic Dynamics of a Semiconductor Nanolaser Subjected to FBG Feedback. Opt. Express, OE 2021, 29, 17815–17830, doi:10.1364/OE.427541.

The following are typos errors:

In Abstract, line 13: “we” should be “We”.

Reply: We thank the reviewer for pointing out this. In the manuscript, “we employ bifurcation diagrams and 0-1 chaos tests in our simulations to quantitatively analyze the dynamic properties of the system” was amended to “We employ bifurcation diagrams and 0-1 chaos tests in our simulations to quantitatively analyze the dynamic properties of the system”.

Page 2: “section *” should be “Section *”

Reply: Thanks. In the manuscript, “In section 2”、“In section 3”、and “in section 4” were modified to “In Section 2”、“In Section 3”、and “in Section 4” respectively.

Page 4, line 122: “We” should be “we”.  

Reply: Thanks. In the manuscript, “we numerically solve Eqs. (1)-(5) using the fourth-order Runge-Kutta method in this study.” was modified to “We numerically solve Eqs. (1)-(5) using the fourth-order Runge-Kutta method in this study.”

Page 4, line 128: “Where” should be “where”.  

Reply: Thanks. In the manuscript, “Whererepresents the average value of the time series” was modified to “where <> represents the average value of the time series”.

Page 5, line 139: “the Figure 2” should be “Figure 2”.  

Reply: Thanks. In the manuscript, “As can be seen from the Figure 2” was modified to “As can be seen from Figure 2”.

Page 5, line 140: “strength” should be “strengths”.  

Reply: Thanks. In the manuscript, “when the injection strength are 13” was modified to “when the injection strengths are 13”.

Page 6, line 165: “Figure 2, 3” should be “Figures 2, 3”.

Reply: Thanks. In the manuscript, “This finding is similar to those observed in Figure 2, 3” was modified to “This finding is similar to those observed in Figures 2, 3”.

Page 7, line 208: “Figure 8, 9” should be “Figures 8, 9”.

Reply: Thanks. In the manuscript, “and the results are shown in Figure 8, 9” was modified to “and the results are shown in Figures 8, 9”.